# Reproducible Propagation of Species-Rich Soil Bacterial Communities Suggests Robust Underlying Deterministic Principles of Community Formation

Senka Čaušević,[a] Janko Tackmann,[b] Vladimir Sentchilo,[a] Christian von Mering,[b] Jan Roelof van der Meer[a]

[a]Department of Fundamental Microbiology, University of Lausanne, Lausanne, Switzerland
[b]Department of Molecular Life Sciences, University of Zürich, Zürich, Switzerland

**ABSTRACT** Microbiomes are typically characterized by high species diversity but it is poorly understood how such system-level complexity can be generated and propagated. Here, we used soil microcosms as a model to study development of bacterial communities as a function of their starting complexity and environmental boundary conditions. Despite inherent stochastic variation in manipulating species-rich communities, both laboratory-mixed medium complexity (21 soil bacterial isolates in equal proportions) and high-diversity natural top-soil communities followed highly reproducible succession paths, maintaining 16S rRNA gene amplicon signatures prominent for known soil communities in general. Development trajectories and compositional states were different for communities propagated in soil microcosms than in liquid suspension. Compositional states were maintained over multiple renewed growth cycles but could be diverged by short-term pollutant exposure. The different but robust trajectories demonstrated that deterministic taxa-inherent characteristics underlie reproducible development and self-organized complexity of soil microbiomes within their environmental boundary conditions. Our findings also have direct implications for potential strategies to achieve controlled restoration of desertified land.

**IMPORTANCE** There is now a great awareness of the high diversity of most environmental ("free-living") and host-associated microbiomes, but exactly how diverse microbial communities form and maintain is still highly debated. A variety of theories have been put forward, but testing them has been problematic because most studies have been based on synthetic communities that fail to accurately mimic the natural composition (i.e., the species used are typically not found together in the same environment), the diversity (usually too low to be representative), or the environmental system itself (using designs with single carbon sources or solely mixed liquid cultures). In this study, we show how species-diverse soil bacterial communities can reproducibly be generated, propagated, and maintained, either from individual isolates (21 soil bacterial strains) or from natural microbial mixtures washed from top-soil. The high replicate consistency we achieve both in terms of species compositions and developmental trajectories demonstrates the strong inherent deterministic factors driving community formation from their species composition. Generating complex soil microbiomes may provide ways for restoration of damaged soils that are prevalent on our planet.

**KEYWORDS** soil microcosms, community development, colonization, mixed bacterial species growth, microbial communities, soil microbiology

Address correspondence to Jan Roelof van der Meer, JanRoelof.VanderMeer@unil.ch.

The authors declare no conflict of interest.

Microbial communities are highly complex systems that self-organize seemingly spontaneously within the spatiotemporal, physical, chemical, and biological boundary conditions of their environment or their host. The living microbial systems

within these boundaries (the "microbiomes") have attracted recent wide interest, due to their crucial contributions to ecological and biosphere processes (1–3), as well as to plant (4), human (5), and animal health (6). However, despite their widely recognized importance, there is still a large gap in understanding the general principles underlying microbiome development and functioning, as well as their amenability for functional and compositional engineering.

To a large part, our current understanding of the operating principles of microbiome formation comes from bottom-up studies with limited species numbers in synthetic ecosystems (7–10). Interspecific interactions are assumed to be the generators of community self-assembly and of emerging system-level metabolic properties (11, 12). For example, range expansion experiments with two to three bacterial strains have demonstrated the quality, types, and importance of interspecific metabolic interactions and spatial structuring (13–18). To some extent, higher-order community composition can also be successfully predicted from empirical measurements of paired growth interactions (10, 19). However, multi-species interactions can give rise to feedback mechanisms that provide reciprocal control on their growth (10), or lead to multistable paths as a consequence of individual growth variation (20). Interspecific interactions further emerge in dependency of initial growth conditions and environments (21, 22), and with increasing species complexity, non-additive effects may arise (23). The emergence of interspecific interactions depends on the spatial distance between cells (24) and, consequently, may be different in highly fractured environments such as soil, as opposed to liquid suspension (25–28). The question is thus whether developmental paths of species-rich communities are inherently stochastic and, in that sense, mostly irreproducible, or whether their taxa-composition provides robust self-organizing properties that will only diverge as a result of differences in environmental boundary conditions. In order to test this question, it is important to design studies that can bridge from the very simplified synthetic bacterial communities alluded to above, to more realistic species-diverse communities.

The major aims of the underlying work were thus 2-fold: first, to develop a tractable system to generate and propagate species-rich communities, and second, to study their developmental paths and resulting compositional states under different environmental boundary conditions and culturing regimes. We specifically focus on soil microbiomes, which comprise among the most diverse known microbial communities with up to 50,000 prokaryotic species (29) and $10^{10}$ cells per gram of material (30). In addition, soils are hosts to multitudes of eukaryotic microbes, including fungi and protists, and phages (1). The soil microbiome is of crucial importance for soil fertility and plant growth, for water purification and biogeochemical cycles (1, 31, 32). Soils are threatened worldwide as a result of land management, agricultural practices, erosion, waste deposition, or chemical spills, leading to a general loss of soil structure and diversity (33, 34). Soil microbiomes are thus highly relevant and one of the options for restoration of perturbed communities is through rational management, although current methods, e.g., soil transplantation or inoculation are very much a black box (35–38).

We contrasted development of two types of soil communities, one composed of 21 indigenous soil isolates covering four major phyla (called *synthetic community* or SynCom), and the other comprising a species-rich soil microbial mixture directly washed and purified from a forest top soil (NatCom, for natural community). Both communities were inoculated at low density in sterile soil matrix under aseptic conditions to allow growth and colonization, under two different culturing regimes (Fig. 1A). The first consisted of a single long-term incubated batch sampled after 1 week, 2 months, and 6 months, to favor slow-growing taxa. The other consisted of multiple dilution-growth cycles of 1 week each, to favor community stabilization and test resilience to chemical perturbation. Community trajectories in the soil matrix were further compared with that in liquid suspension. As our main focus was the bacterial communities, we inferred compositional changes from 16S rRNA gene amplicon sequencing, while being aware that this neglects eukaryotic microbes or phages that may have been

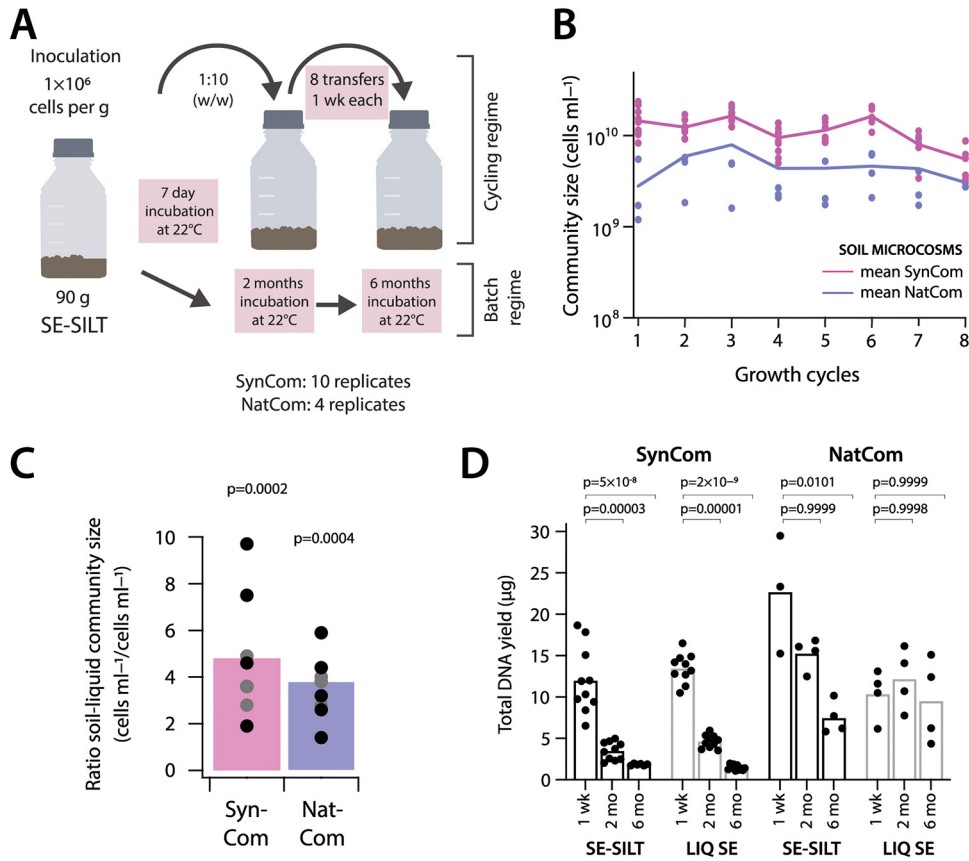

**FIG 1** Development of synthetic and natural soil microbial communities in soil microcosms. (A) Freshly washed soil communities (NatCom) or synthetic composed soil bacterial community (SynCom, 21 species) were used to inoculate four and 10 replicate sterile soil microcosms (each 100 g soil, $10^6$ cells g$^{-1}$ at start), respectively. Microcosms were incubated for 7 days and then diluted into sets of fresh sterile microcosms (1:10, wt/wt). This growth cycling was repeated for a total of eight cycles. For the long incubations, the same initial microcosms were sampled after 1 week, 2 months, and 6 months. (B) Sizes (in cells mL$^{-1}$ soil liquid phase, determined by flow cytometry) of NatCom (cyan) and SynCom communities (magenta) across eight subsequent growth cycles. Lines connect the mean cell count of all replicates (NatCom: four replicates, SynCom: 10 replicates, except after the fifth cycle where five replicates were removed for exposure to toluene) at the end of each transfer, with dots indicating individual values. (C) Mean (bars) and individual (dots, gray to black shades) for ratios of SynCom (magenta) and NatCom (cyan) flow cytometry cell counts after each growth cycle in SE-silt compared with suspended growth in liquid SE. P-values refer to one-tailed paired t test of SE-silt values versus liquid SE suspensions. (D) Mean (bars) and individual (dots) replicate DNA yields from SynCom and NatCom communities after 1 week, 2 months, and 6 months incubation in SE-silt or in suspended growth in liquid SE (LIQ SE). P-values refer to one-tailed paired t-tests in comparison to the 1–week DNA yields of the same sample group, with the *alternative* hypothesis that values at later time points are lower than week 1. SE, soil extract.

present in the NatComs. Community composition signatures were compared with all available worldwide soil and rhizosphere communities characterized by 16S rRNA gene amplicon sequencing. Our results indicate highly reproducible bacterial community development for both synthetic and species-rich natural soil inocula. Developmental trajectories depend on incubation regimes and environmental conditions, suggesting robust deterministic self-organizing principles.

## RESULTS

**Generation of controllable soil microbiome culturing systems.** Standardized solid-phase culturing systems for studying the development and succession of species-rich microbial communities were produced from twice-autoclaved silty soil originating from a riverbank (39). The silt matrix itself has low total organic matter content (0.13%) (39), to which was added a sterile liquid soil nutrient extract (soil extract [SE]), in order to provide complex organic nutrients from the same soil as used to extract the washed NatCom cells (see Materials and Methods; SE and silt characteristics in Fig. S1). This

**TABLE 1** Taxonomy of selected strains for the synthetic soil community (SynCom)

| No. | Genus | Class | Phyla |
|---|---|---|---|
| 1 | Microbacterium | Actinobacteria | Actinobacteria |
| 2 | Mucilaginibacter | Bacteroidia | Bacteroidetes |
| 3 | Curtobacterium | Actinobacteria | Actinobacteria |
| 4 | Variovorax | Gammaproteobacteria | Proteobacteria |
| 5 | Flavobacterium | Bacteroidia | Bacteroidetes |
| 6 | Cellulomonas | Actinobacteria | Actinobacteria |
| 7 | Tardiphaga | Alphaproteobacteria | Proteobacteria |
| 8 | Devosia | Alphaproteobacteria | Proteobacteria |
| 9 | Mesorhizobium | Alphaproteobacteria | Proteobacteria |
| 10 | Burkholderia | Betaproteobacteria | Proteobacteria |
| 11 | *Pseudomonas* | Gammaproteobacteria | Proteobacteria |
| 12 | Luteibacter | Gammaproteobacteria | Proteobacteria |
| 13 | Chitinophaga | Bacteroidia | Bacteroidetes |
| 14 | Lysobacter | Gammaproteobacteria | Proteobacteria |
| 15 | *Pseudomonas* | Gammaproteobacteria | Proteobacteria |
| 16 | Rhodococcus | Actinobacteria | Actinobacteria |
| 17 | Caulobacter | Alphaproteobacteria | Proteobacteria |
| 18 | Cohnella | Bacilli | Firmicutes |
| 19 | Rahnella | Gammaproteobacteria | Proteobacteria |
| 20 | Phenylobacterium | Alphaproteobacteria | Proteobacteria |
| 21 | Bradyrhizobium | Alphaproteobacteria | Proteobacteria |

nutrient-complemented material (referred to as "SE-silt" and "soil microcosms" for the rest of the manuscript) was filled in flasks that could be inoculated, grown and diluted into fresh material, as is common for typical liquid culturing under aseptic conditions (Fig. 1A). Organic matter analysis of SE-silt indicated an average of 1.5 mg total organic carbon $g^{-1}$ soil matrix and 0.3 mg total N $g^{-1}$ (Table S1). Assuming carbon needs of 200 fg C per cell and a g-C g-C$^{-1}$ yield ratio of 20%, this would permit the development of a community of roughly $10^9$ cells $g^{-1}$, which is similar to the measured microbial community size in the silt from total cell counts (39).

The material was inoculated with starting community suspensions at $10^7$ cells mL$^{-1}$, estimated from flow cytometry counting; producing an equivalent of $10^6$ cells $g^{-1}$ soil at the set 10% gravimetric water content. Community inocula consisted either of a washed and purified microbial cell suspension from top-soil (NatCom) or a mixed suspension of 21 soil bacterial isolates (SynCom, see below, Table 1). Inoculated soil microcosms with NatCom suspensions after 1 week reached $2.8 \pm 2.4 \times 10^8$ cells $g^{-1}$ (one *SD*, $n = 4$, Fig. 1B), an estimated 280-fold increase from the inoculum size (~8 doublings). Averaged across all 1-week culturing cycles, the NatComs maintained at $4.7 \pm 1.1 \times 10^8$ cells $g^{-1}$ material. This was an average of 3.5 times higher than the community size obtained in (liquid) SE solution alone (calculated on a per mL–basis, Fig. 1C; $P = 0.0004$, one-tailed *t* test). This suggested that all easily accessible carbon was utilized during each week of incubation time and that communities reached semi-stationary phase (see below) before they were transferred to fresh soil microcosms. There was no discernible trend in the NatCom cell numbers as a function of growth cycle (Fig. 1B, NatCom linear regression: 0.0211, $P = 0.4371$ compared with slope = 0). SynCom inoculum mixtures (Table 1) increased from $1 \times 10^6$ cells $g^{-1}$ material to a stable average density after every growth cycle of $1.11 \pm 0.32 \times 10^9$ cells $g^{-1}$, which was 2.4 times higher than that of the NatCom (Fig. 1B; $P = 9.9 \times 10^{-9}$ unpaired two-sided *t* test, $n = 34$). In comparison to its liquid SE suspension, the average 1-week SynCom density in SE-silt was four times higher, similar as for the NatCom (Fig. 1C; $P = 0.0002$, one-tailed *t* test).

In contrast to the communities propagated under the 1-week soil microcosm growth/dilution cycles, those maintained under a single long batch incubation decreased in size from the first week to two and 6 months, as inferred from community DNA yields (Fig. 1D). NatCom DNA yields in soils decreased by 2- to 4-fold after 2 and 6 months, but not in liquid SE (Fig. 2C; and $P = 0.0101$). SynCom sizes declined by 3-

and 6-fold after 2 and 6 months, respectively, both in soils and liquid (Fig. 1D). This decrease may have been due to carbon limitation, consequent cell death and carbon turnover, or predation. Overall, these experiments indicated that high-density complex communities developed in both regimes and persisted over long times.

**Compositional state trajectories during culturing.** The NatCom compositional dynamics under the two growth regimes was assessed from changes in the relative taxa abundances, determined by 16S rRNA gene amplicon sequencing using 99% identity thresholds for operational taxonomic unit (OTU) assignment. Although this neglects possible eukaryotic microbes or phages that may have been present in the originally extracted NatCom cell suspension, this restriction is justified for ease of comparison to the exclusively bacterial SynComs (see below). The mean detected richness reduced from 233 in the inoculum to 22 (9%) after the first week, which slowly increased to 37 (16%) after the eighth incubation cycle (Fig. 2A). In addition, 75% of the taxa after the first cycle (week 1) were not detected in the inoculum (Fig. 2A, magenta bars), suggesting that community succession was initially driven by rapidly growing low abundant taxa. Non-metric multidimensional scaling (NMDS) analysis confirmed the strong deviation of both SE-silt and liquid SE microcosms from that of the original inoculum (Fig. 2B). Growth cycles resulted in closely clustering communities (Fig. 2B, T1 to T8), whereas the single long incubations showed succession, higher richness (78.5 $\pm$ 9.5 OTUs after 6 months, Fig. 2A) and higher similarity to the inoculum state (Fig. 2B, 2 and 6 months). NatCom development in SE-silt was distinct from that in liquid SE alone, indicating that the soil microcosm environment may have driven the community differentiation (Fig. 2B, adonis $P = 0.001$ with beta-dispersion of $P = 4.38 \times 10^{-10}$).

Although replicates clustered coherently in NMDS (Fig. 2B), there were obvious stochastic effects of compositional succession, illustrated by variation in appearance and relative abundance of individual taxa among replicate inoculations after the first week of incubation (Fig. 2C and e.g., Rhizobiales, Sphingomonadales, and Enterobacteriales). Replicate variability was higher at OTU level than at order level (Fig. 2D, Fig. S2), suggesting conserved functional order traits that permit strains from such groups to quickly colonize new environmental niches. NatCom replicates kept a relatively strong individual signature independent of multiple growth/dilution cycles (most evident with the "−2" replicate, Fig. 2E), which mostly converged in long-term incubations (Fig. 2E, L6 samples). This might be due to stochastic variations of species composition in the subsamples of the inoculum mixture used to start the NatCom replicates, which then influence growth in the first incubation and from there on, propagate the states of regrown communities. Mathematical simulations of community growth and composition suggested that this variation may be due to subsampling effects of rare taxa with high growth rates within a finite-sized inoculum (Fig. S3). Initially composed of 18 phyla, only five were detected in NatCom replicates after the first growth cycle, and four more appeared after cycle 8 (Fig. 2F), indicating that their members were present but undetectable at our sequencing depth. In contrast, long-term incubated NatCom showed members of 10 phyla, indicating that this growth regime permitted higher diversity, perhaps by avoiding bottlenecks of the dilution/growth cycles on slow-growing members (Fig. 2F). This showed that species-rich communities can be grown and maintained with relatively constant composition over multiple dilution cycles, despite having inter-replicate stochastic strain variability. Culturing in soil microcosms clearly provided additional benefits to the community, since both its size (Fig. 2B) and its richness remained larger (by 12.02% with growth cycles and 9.31% in the long batch regime, Fig. S4) than that in SE liquid suspension.

**Development of medium complexity synthetic soil microbiome recapitulates natural states.** To place the observed succession and development patterns in the NatComs in perspective, we compared them with community development of a defined SynCom under the same growth conditions. The SynCom was composed of 21 bacterial isolates that were selected from a total of 169 recovered pure cultures from the same soil as the NatCom (see Materials and Methods). Pure cultures were first selected based on different colony morphologies and growth characteristics, and

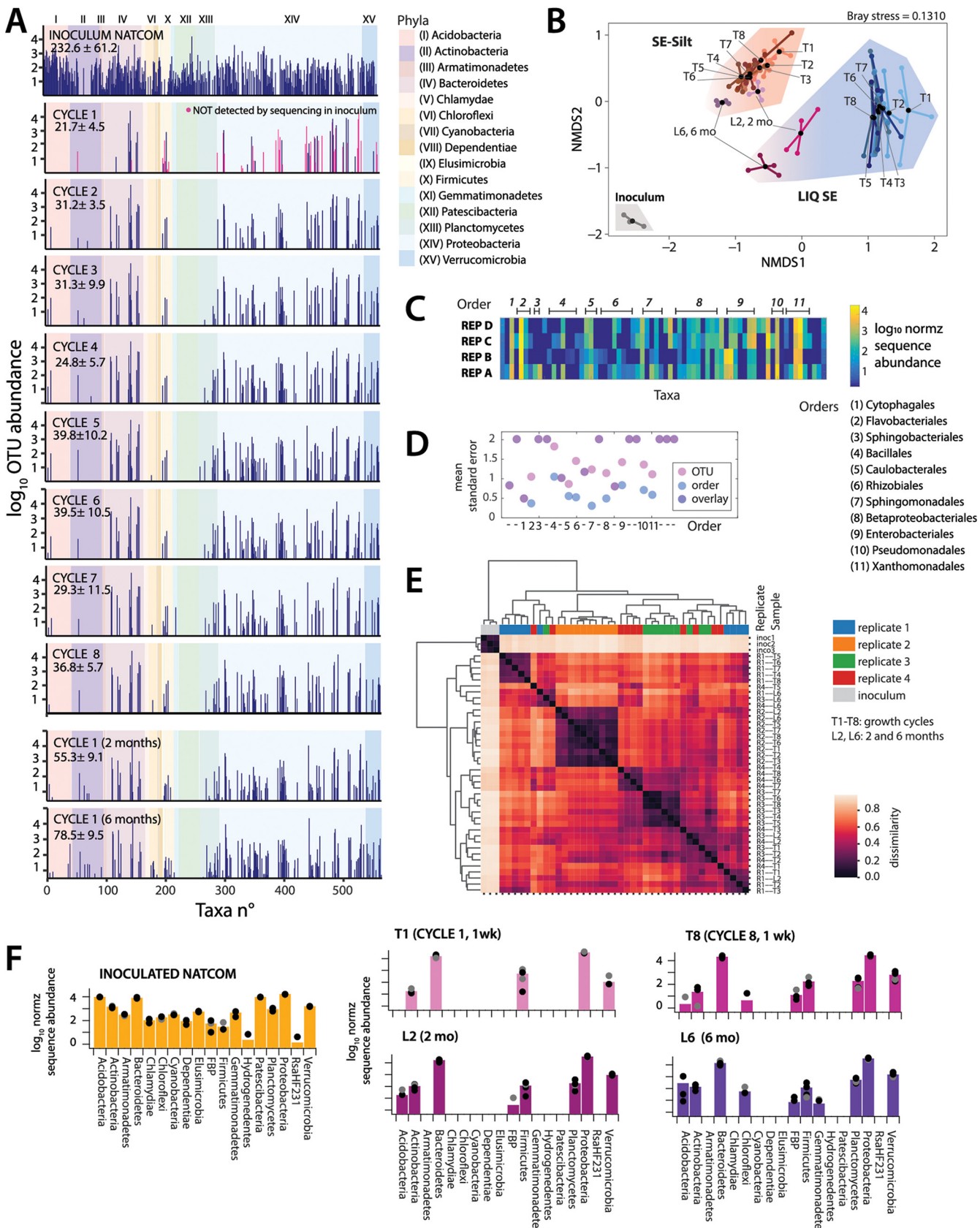

**FIG 2** Community succession and composition of NatComs. (A) Mean log$_{10}$-transformed total read-normalized (5 × 10$^4$) taxa abundances in the soil inoculum, after all the eight 1-week growth cycles, and in the long incubation time points (2 and 6 months). Abundance bars positioned according to taxa

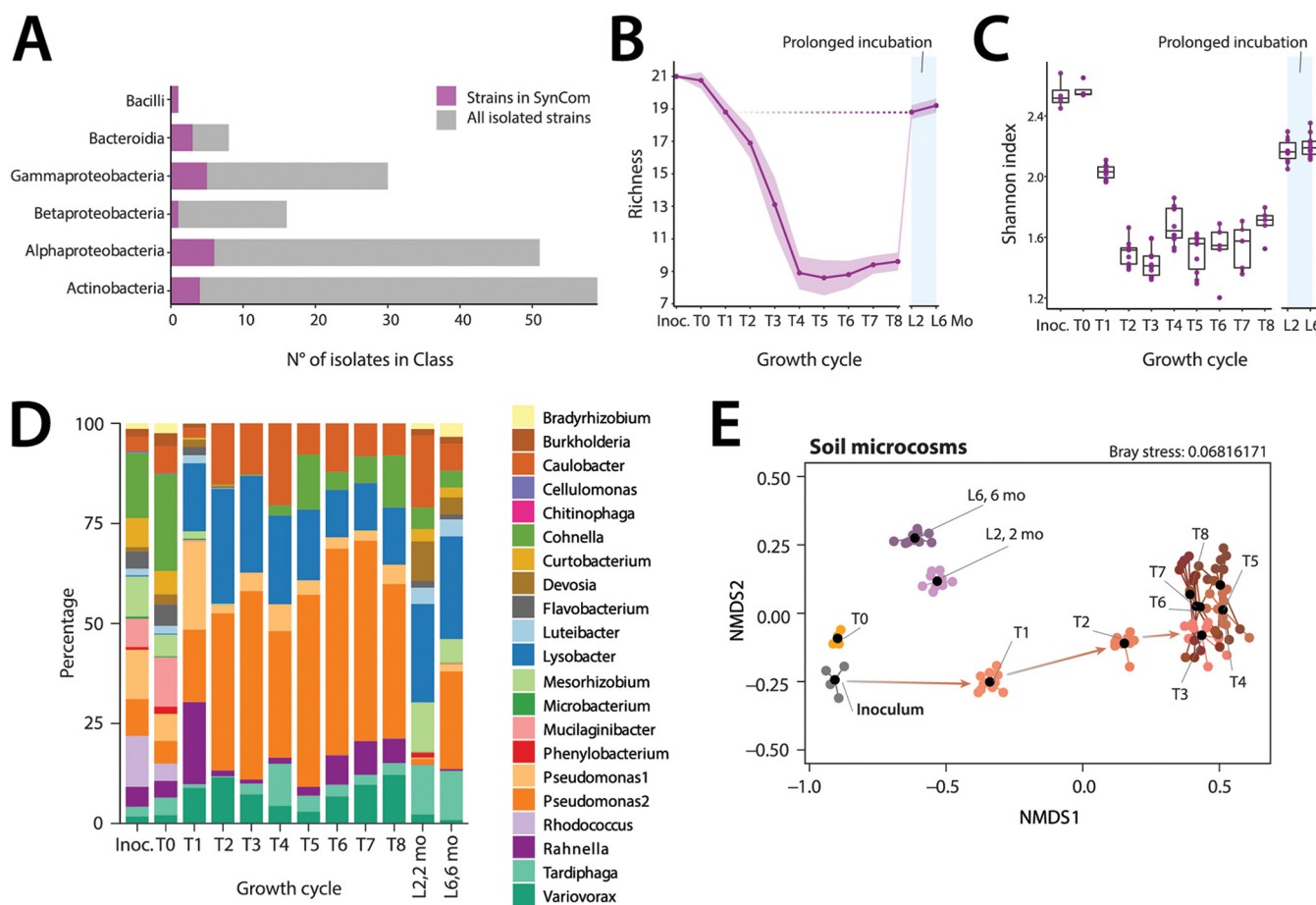

**FIG 3** Succession and stabilization of a synthetic soil community over multiple growth cycles and long-term incubation. (A) Class attribution of the 172 isolated soil bacterial strains, and of the selected 21 strains of the SynCom. (B) Changes in mean SynCom richness (magenta line, ± one *SD* in light color background, *n* = 10 replicates) and (C) in mean Shannon indices (box plots, *n* = 10; except T6-T8; *n* = 5 replicates) throughout the eight growth cycles in SE-silt (T1 to T8), and during long-term incubation (L2, L6; 2 and 6 months, mo). (D) Stacked mean relative abundances (in percentage, *n* = 10 replicates) of SynCom members (legend on the right) from inoculation to the last growth cycle, and upon long-term incubation. (E) Non-metric multidimensional scaling of normalized SynCom compositions in soil microcosms according to their Bray-Curtis distances. Black dots show the community centroids; colored dots are individual replicates.

covered 52 different genera belonging to four phyla on the basis of their 16S rRNA gene sequences (Table S2). Not surprisingly, despite trying different culture media and growth conditions, this isolation resulted in a reduced representation of the NatCom (Table 1). We therefore based the SynCom choice of 21 isolates on a diverse selection of major phyla observed in the NatCom after the first soil microcosm growth cycle (Actinobacteria, Bacteroidetes, Firmicutes, and Proteobacteria, Fig. 2F), including some taxa redundancies (Table 1, Fig. 3A). All isolates were then cultured individually and mixed in equal proportions before inoculation into the microcosms.

**FIG 2** Legend (Continued)

numbering from the OTU list (SILVA, at 99% similarity), with background color representing phyla affiliation (Roman numbering, according to legend). Numbers within panels show mean taxa richness ± one *SD* (*n* = 4 replicates). Magenta bars in the CYCLE-1 data point to taxa not detected in the inoculum. (B) Non-metric multidimensional scaling ordination of NatCom succession in SE-silt (magenta area), or in liquid SE suspension (cyan area; T1 to T8, weekly transfers; L2, L6, 2 and 6 months incubations). Ordination plot based on Bray-Curtis distances. (C) Compositional variation (shown as $\log_{10}$-normalized abundance heatmap) among the four NatCom replicates (REP A to D) after the first growth cycle. Numbers above refer to taxa within order-levels as specified on the right. (D) Mean standard error of replicate variation (REP A to D after 1 week) at OTU-level (mean of means grouped within corresponding order, pink) or at order-level (blue; purple is where both OTU- and order-values overlap). Note how order-level variation is lower than OTU-variation. Numbers refer to order in (C). -, single OTU in order; not specified. (E) NatCom pairwise sample comparison, clustered by average-linked Bray-Curtis distances (color scale). Inoculum (soil 1 to 3) and replicates (R1 to 4) are highlighted by different colors on the top, growth cycles (T1 to T8) or long-term incubations (L2, 2 months; L6, 6 months) in small fonts on the right. Note the strongly maintained replicate signatures (e.g., replicate 2). (F) Mean (bars) and individual replicate (gray to black dots, *n* = 4) grouped phyla composition of NatCom inoculum (orange), after the first growth cycle (CYCLE 1, 1 wk), the eighth (CYCLE 8), and after 2 and 6 months (mo).

In contrast to NatCom, the compositions of the SynCom needed three growth/dilution cycles before stabilizing. Succession was evident from a loss of apparent diversity (i.e., within the sequencing threshold for community membership), from 21 to nine to 10 detectable members after the fourth cycle (Fig. 3B), and a sharp decrease of Shannon index (Fig. 3C). The $T_0$–sample (taken 30 min after inoculation into the soil) resembled the inoculum closely (Bray-Curtis distances of 0.26 $\pm$ 0.02, while the distance between inoculum and T8 was 0.65 $\pm$ 0.03), showing minimal bias introduced by cell extraction (Fig. 3D). Initially higher relative abundances of *Pseudomonas* strain1 and *Rahnella* during the first-to-third growth cycles were replaced by *Pseudomonas* strain 2, *Lysobacter*, *Variovorax*, and *Caulobacter* as the dominant members. Finally, also *Cohnella*, *Rahnella*, and *Tardiphaga* regained sizeable proportions of the SynCom (Fig. 3D). Independent SynCom replicates followed highly similar developmental paths (Fig. 3E), in terms of compositional changes, loss of diversity and reaching semi-stable compositions after the fourth cycle (Fig. 3B to D). SynCom replicates clustered coherently over time and did not maintain individual replicate signatures as NatCom (Fig. S4). SynCom compositions in soil microcosms differed significantly from that of the inoculum and those grown in liquid SE suspension (Fig. S5; adonis $P$ = 0.001; betadisper $P$ = 0.0002). Similar to NatCom, the long incubation regime led to higher detectable diversity of 18–20 (out of 21) strains after 2 and 6 months (Fig. 3B and C, and E, $P$ = 0.001 from adonis and $P$ = 0.0002 for beta-dispersion). This included higher relative abundances of *Mesorhizobium*, *Luteibacter*, and *Devosia* compared with e.g., *Pseudomonas* (Fig. S6).

**SynCom and NatCom retain soil community signatures but differ in replicate variability.** In comparison with a wide set of publicly available data sets ($n$ = 110,928) on soil communities characterized by 16S rRNA gene amplicon sequencing, both SynCom and NatCom compositions grown in soil microcosms kept clear soil community signatures (Fig. 4A). Interestingly, SynCom compositions located closer to "plant rhizosphere" communities, possibly due to the culturing isolation bias (Fig. 4A). NatCom grouped closer to "field soils," whereas the inoculum, as expected, had a "forest" soil signature (Fig. 4A). There is not a clear single factor underlying this environmental signature, although soil-pH (as far as present in the meta-data) seems an important variable (Fig. 4B). Both SynCom and NatCom became largely dominated by Alpha- and Gammaproteobacteria, but were notably different in the relative abundances of Bacteroidetes (contributing 30% to 50% in the NatCom) and Firmicutes (5% to 10% in the SynCom) (Fig. 4C). SynCom replicate variability was twice as low as that of the NatCom (Fig. 4D and $F$ = 17.495, $P$ = 5.19 $\times$ 10$^{-5}$, ANOVA), with high replicate homogeneity (i.e., the replicate Bray-Curtis distance from the community centroids, ranging from 0.01 to 0.20; Fig. 4D). The reason for this is likely the lower number of starting strains in the SynCom and the lower likelihood of stochastic variations as a result of subsampling upon dilution (as in, e.g., Fig. S3).

**Chemical perturbation changes SynCom trajectories.** In order to investigate the stability of developed communities, we tested their resilience toward the moderate toxic compound toluene, as an example of recurrent soil pollution with organic solvents (40). To this end, we split the stabilized 10 SynCom replicates in two groups of five after the fifth growth cycle; one series of which was exposed to toluene vapor during the next 1-week cycle, the other cultured as before. After this exposure period, all SynCom replicates were diluted again into sterile, non-polluted soil microcosms; and growth cycles were continued as before. Exposure to toluene significantly lowered the attained community sizes (Fig. 5A and $P$ = 0.0231, one-tailed $t$ test on all replicate samples and time points, $n$ = 15). In contrast, toluene exposure did not lead to significant changes in richness (Fig. 5B and $P$ = 0.4235, two-way ANOVA), nor did it influence Shannon diversity (Fig. 5C and $P$ = 0.2128, two-way ANOVA). Varying effects were observed on individual SynCom members, which either slightly (Fig. 5D and e.g., *Variovorax*, 25.8% decrease), or drastically decreased in population size (Fig. 5D, 99.9% decrease of *Burkholderia*, see *Devosia* and *Flavobacterium* in Fig. S7), whereas some increased in abundance (Fig. 5D, *Microbacterium*, *Cohnella*). Inter-replicate variability

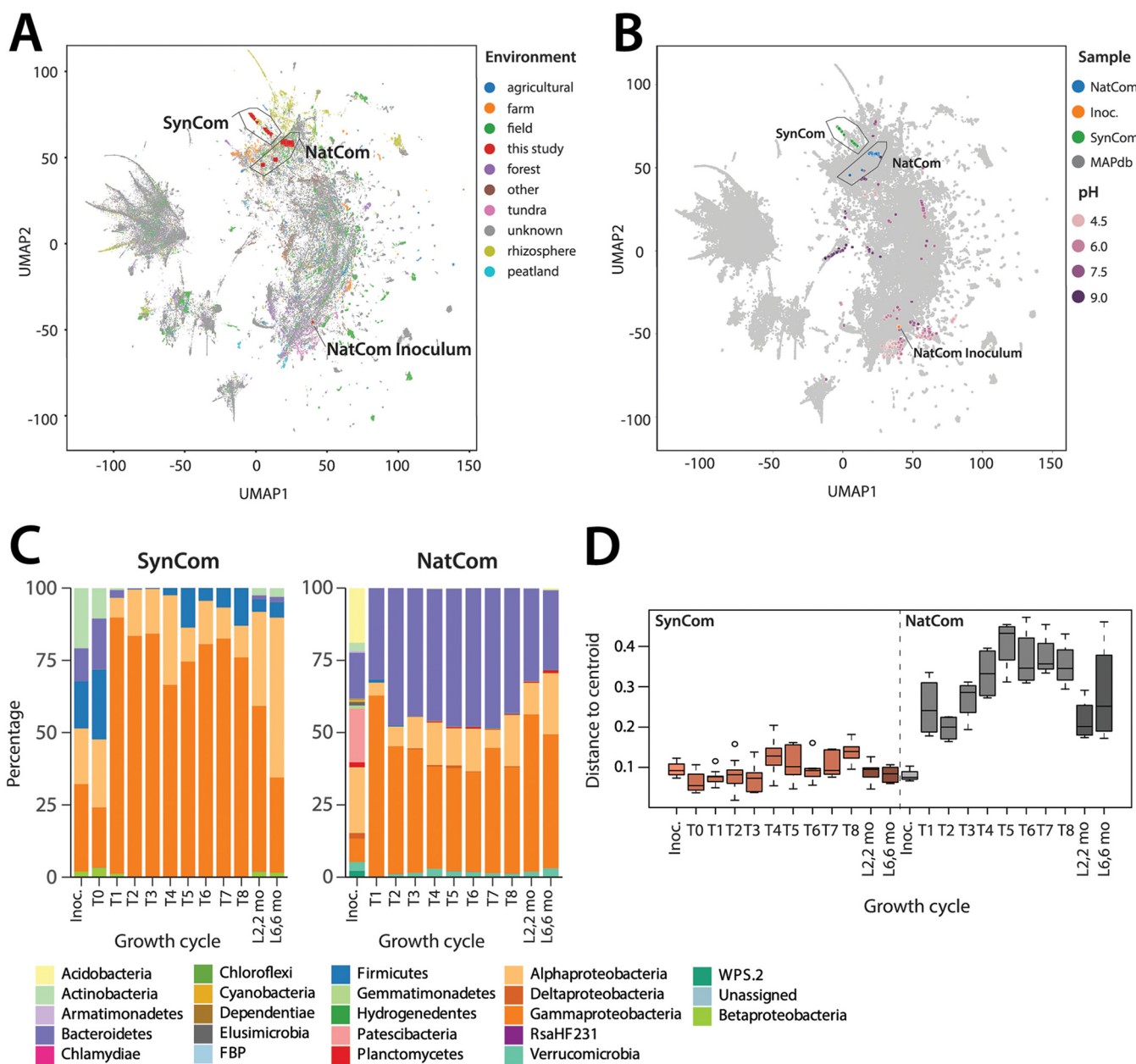

**FIG 4** NatCom and SynCom community signatures. (A) Environmental signature of NatComs and SynComs. Map shows a UMAP projection of SynCom and NatCom samples together with 110,928 soil communities (dots) extracted from the Microbe Atlas Project (71), based on Bray-Curtis distances and color coded along their environmental origin, or (B) overlaid with soil pH, extracted from the Earth Microbiome Project (72). (C) SynCom and NatCom relative abundances at phyla and class levels (Proteobacteria only). (D) Interreplicate variability of SynCom and NatCom replicates, shown here as individual Bray-Curtis distances to the corresponding community centroid. Boxplots show 25th, median and 75th percentiles, with whiskers indicating 1.5× the interquartile range.

was not significantly affected with toluene exposure, even during the first week of recovery (Fig. 5E and $F = 0.8973$, $P = 0.4994$, ANOVA). Community signatures in exposed SynCom remained distinct from those of the non-exposed communities even after the 8th cycle (Fig. 5F, adonis $P = 0.001$; betadisper $P = 0.1024$). Altogether, this indicated that chemical perturbation by toluene exposure, changed SynCom compositional trajectories in a long-lasting manner.

## DISCUSSION

We showed reproducible assembly, succession. and composition of both a high-complexity NatCom (starting from washed mixed soil inoculum, containing 18 bacterial

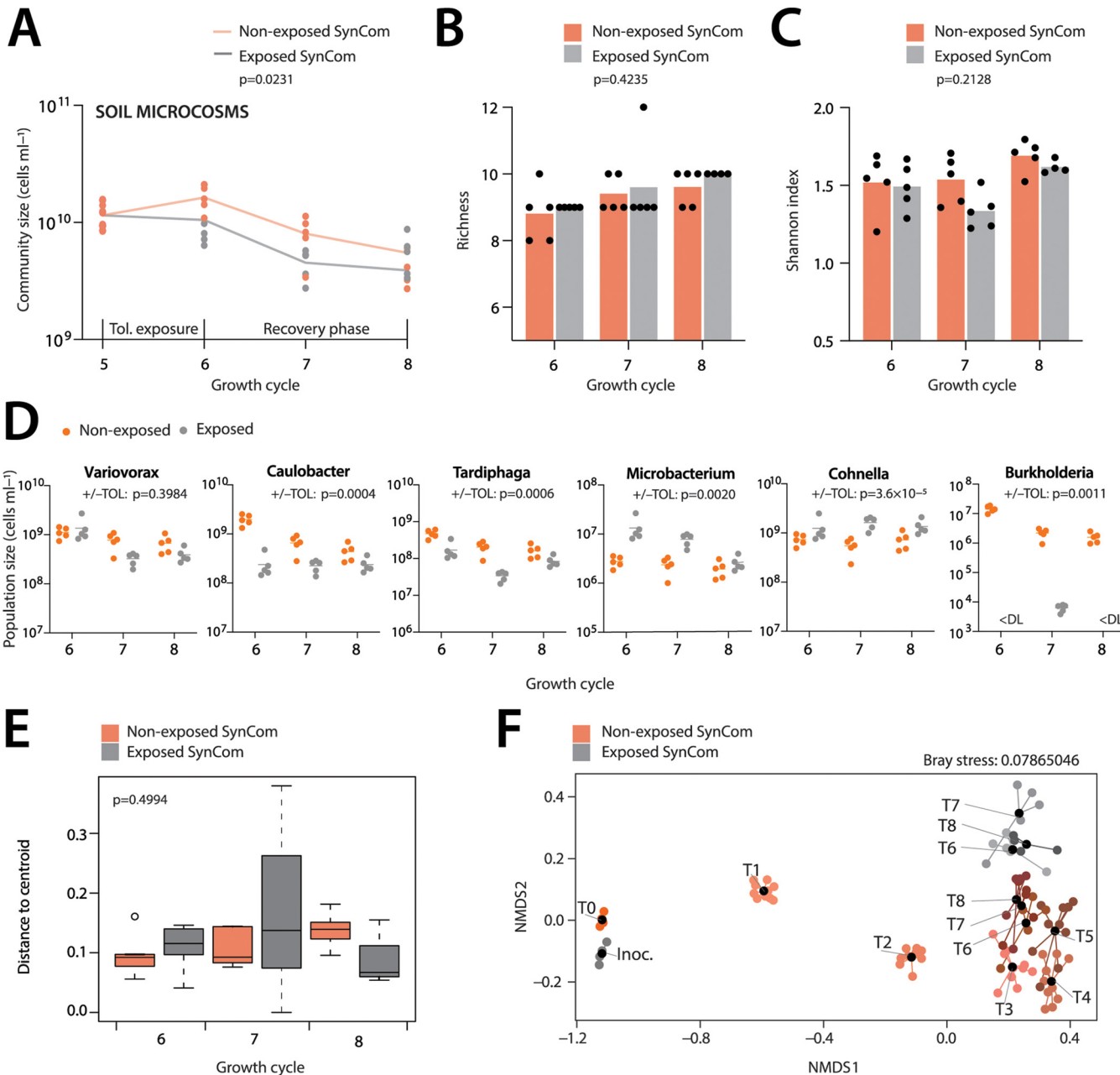

**FIG 5** Community resilience upon chemical perturbance. (A) Mean (lines) and replicate (dots) SynCom size changes (in cells mL⁻¹ soil liquid phase) in toluene-exposed (gray) versus non-exposed (orange) microcosms (each five replicates). Toluene exposure during 1 week of the fifth growth cycle. P-value refers to comparison of cell densities after the sixth to eighth cycles between exposed and non-exposed communities in a Wilcoxon matched-pairs signed rank test. (B) Mean (bars) and replicate (dots) richness in toluene-exposed versus non-exposed SynComs. P-values from two-way ANOVA for toluene exposure. (C) as B but for Shannon index. (D) Changes in absolute abundances (calculated from individual relative sequence abundances and total community size by flow cytometry) of selected SynCom members with and without toluene exposure (two-sided t test, grouped T6 to T8 values). (E) Interreplicate variability, expressed as average distance of replicates to community centroid. P-value from ANOVA of exposed versus non-exposed centroid distances. (F) Toluene-exposure effect on SynCom compositions after the sixth to eighth growth cycles (NMDS based on Bray-Curtis community distances). Sample abbreviations (e.g., T1) as before; black dots, community means; gray dots, toluene-exposed SynCom replicates during T5; orange to brown dots; non-exposed SynCom.

phyla), as well as a medium-complexity SynCom with 21 culturable strains (covering four major bacteria phyla) in a soil-matrix culturing system that enables easy solid-phase transfers. Although we restricted ourselves here to analysis of the bacterial taxa, both NatCom and SynCom retained typical soil and plant rhizosphere bacterial community signatures. They may thus represent excellent test beds for plant growth, community management or soil resilience studies that require complex and reproducible

starting communities. Both growth regimes, either imposed as multiple 1-week growth and dilution cycles in soil microcosms, or as single batch long incubation (up to 6 months), favored establishment of high species diversity, which in short incubations (1 week) was dominated by relatively fast-growing opportunistic strains. Cultured SynCom on average had higher cell densities on the same substrate than NatCom, which might be due to their inoculum being exclusively of bacterial origin, without any heterotrophic fungi, potential phage, or protist predators that might have been present in the washed mixed microbial top-soil NatCom inoculum mixture. The sizes of both communities, however, moderately decreased over long incubation periods (2 and 6 months), suggesting some cell death or predation (even SynComs contained a potential opportunistic bacterial predator in form of *Lysobacter* (41), and consequent carbon turnover. The long incubation time may have allowed growth of members from slow-growing phyla that are difficult to obtain in pure culture (e.g., Acidobacteria, Gemmatimonadetes).

One of the key surprises of our work is the demonstration of highly reproducible trajectories and compositional states of medium-to-high species-diverse soil communities. Having low variability replicate communities may make it easier to detect effects of inoculant bacteria in relation with plants for improving plant health (42), to investigate the influence of bioaugmentation agents in pollution removal (35, 36), or to address further ecological questions on community resilience, species redundancy or invasion resistance (43, 44). The high reproducibility of community development in the soil microcosm culturing conditions was counterintuitive. Considering the complexity of the provided nutrients, the highly fractured porous environment and species-diverse inocula, we expected that stochastic variations in experimental manipulations would lead to chaotic system behavior. Contrary to this intuition, the 21-member SynCom developed high reproducibly among 10 replicates, with similar succession patterns, total community sizes, and relative species abundances. Initially (inoculated) balanced species proportions were quickly replaced by coherent compositional trajectories and states during multiple growth/dilution cycles. The attained SynCom compositional states were dependent on the type of incubation regimes (growth cycles versus one-batch long time), and specific environments (soil microcosms or liquid SE) but retained no individual replicate signatures. Compositional states could be perturbed by short-term chemical exposure, which then reproducibly continued on slightly different trajectories. In contrast, NatCom compositions showed more stochastic variation among replicates, and individual replicate signatures were retained to some extent in the growth/dilution cycles. Community growth simulations suggested that the reason for increased stochastic replicate variability may lie in population bottlenecks arising from finite-sampling of high diverse community inocula with rapidly growing colonizers, leading to a state which then self-propagates in subsequent growth/dilution cycles. We cannot exclude, however, that eukaryotic microbes or phages have played additional roles in shaping NatCom compositions. Despite this, NatCom replicate variation collapsed at a higher phylogenetic level, suggesting similar functional and redundant properties in the complex starting inoculum that are selected during colonization of pristine growth environments. Long-time incubations also dissipated NatCom compositional variations to a large extent. Both SynCom and NatCom in soil microcosms developed and maintained bacterial compositions that matched known soil bacterial community signatures very closely. This indicates strong deterministic influences of the initial species composition on the community development trajectories (i.e., self-organizing complexity) within its system boundaries and the prevailing environmental conditions.

Several authors have reiterated that the origins of microbiome complexity remain fundamentally unknown (23, 45) and that general rules governing community assembly and functioning are difficult to deduce (21, 46–48). It is clear that community growth and development are influenced by a myriad of factors such as growth substrates, spatial structures, and presence of other chemical compounds (45, 49). The

more complex carbon substrates deployed in this study possibly require and facilitate a wider range of metabolic capacities and therefore maintained higher functional diversity (30 to 80 OTUs in NatComs), than in previous experiments starting with soil and phyllosphere communities but grown on a single carbon substrate (5 to 12 exact sequence variants) (50). Community development is further expected to be dependent on emerging interspecific interactions leading to transcending systems-level functionalities (21, 46). Indeed, both NatCom and SynCom development seemed strongly determined by their starting taxa compositions, on top of which the environmental boundary conditions (i.e., soil versus liquid) influencing the community trajectories. The difference in bacterial community compositional trajectories and states in soil microcosms and mixed liquid suspension, despite containing the same soil extract, may be due to different types or magnitudes of interspecific interactions arising in the spatially structured, disconnected and heterogenous growth environment of the soil as opposed to the liquid-suspended growth (25). Soils are expected to provide unique ecological niches (1, 51), and their aggregates affect nutrient availability and gradients in electron donors and acceptors (26–28, 49, 52). Indeed, SynComs and NatComs maintained on average higher species diversity in soil microcosms than equivalent liquid cultures, suggesting emerging favorable dependencies, which permitted more phyla to sustain and grow (25). Suggestive for this is that members belonging to the Acidobacteria, Verrucomicrobia, and Planctomycetes proliferated in all NatCom microcosms, whereas we did not manage to culture them individually using the same nutrient substrates. Despite this, cultured NatComs still lost bacterial diversity compared to the initial inoculated mixture, indicating that other biological factors and interactions, growth conditions, or nutrient sources may need to be present to achieve higher taxa diversity.

Natural soil communities probably only very rarely have the opportunity to colonize a pristine soil environment, except perhaps for soil transplants or soil construction work, grubbing, glacier retreats, or other (53, 54). At a large scale (cm to m), the composition of complex natural soil communities is stable, but may undergo temporal and very local fluctuations driven by nutrient gradients from plant roots, burrowing fauna, rainfall, seasonal temperature changes, or other (55–57). In that sense, our long-term incubation regimes resembled new soil colonization events, eventually leading to a mature state composition, typically comprised of several abundant members and a vast fraction of extremely low abundant species ("rare biosphere") (58–60). The regime of imposed growth cycles may reflect what happens at sudden bursts of newly available carbon in the soil. As the NatCom experiments demonstrated, some "rare taxa" in the mature compositional state as isolated from the natural soil (Gammaproteobacteria, known generalists) rapidly proliferated in the first week of incubation, with Alphaproteobacteria and other phyla appearing only later, as has been observed before in natural systems (48, 60). Some rare taxa may thus rather represent "conditionally rare taxa"; those with radically changing abundances depending on space and nutrient availability (61). The specific roles or capacities of those taxa to become more abundant over time remain unclear for now, and could be due to factors such as use of different (refractory) carbon substrates, predatory lifestyles, different nutrient requirements, or forms of metabolic dormancy to remain viable for longer. From an engineering perspective the maintenance of temporary community compositional steady states by the cycling growth-dilution regime is interesting and suggests an avenue for approaches that aim to keep relatively constant species proportions in mixed communities over time. Reproducible propagation of soil communities will also be key for restoration efforts on degraded or desertified land that aim to bring back healthy soil life.

## MATERIALS AND METHODS

**Preparation of a natural soil community.** A natural mixed microbial community (NatCom) was washed from batches of 20 g taken with a sterile metal spoon from the 5-cm topsoil layer after removal of twigs, roots, and leaves (Dorigny forest, University of Lausanne, 46°31'16.4"N 6°34'43.0"e). Soil batches were immediately transported to the lab and processed within 1 h. The soil was sieved through a 3-mm

mesh to remove large particles. Microbial cells were detached from soil particles by mixing with sterile 0.2% (*wt/vol*) tetrasodium-pyrophosphate decahydrate solution (pH 7.5, Sigma-Aldrich), and then purified by sucrose gradient solution centrifugation as described by (62). The cell suspension recovered after sucrose gradient centrifugation was twice washed with sterile saline solution (0.9% NaCl) and resuspended in the same. Serial dilutions were stained with SYBR green I and cell numbers were counted using flow cytometry (see below). For inoculation into microcosms, the cell suspension was diluted in soil extract (SE, see below) to $10^7$ cells mL$^{-1}$. Subsamples of the NatCom suspension were used for DNA extraction and 16S rRNA gene amplicon sequencing (see below).

**Preparation of the synthetic soil community.** Individual soil isolates were obtained from similar NatCom suspensions of the same soil location, additionally purified using Nycodenz gradient (62), diluted and plated on different media, as suggested by Balkwill and Ghiorse (63). We used PTYG medium (containing, per L: 0.5 g glucose, 0.5 g yeast extract, 0.25 g peptone, 0.25 g Trypticase, 0.6 g MgSO$_4$·7H$_2$O, 0.07 g CaCl$_2$·2H$_2$O, 15 g agar), or soil extract medium (see below) solidified with 1.5% agar (Agar bacteriological, Difco), either at pH 4.5 (adjusted with hydrochloric acid) or at pH 7.5 (with sodium hydroxide). All plates were incubated at room temperature (23°C) for 2 weeks. In total, 169 morphologically distinguishable colonies were selected, purified to homogeneity by streaking on the same medium, regrown in PTYG, and stored in 15% (vol/vol) glycerol at −80°C. Strains were identified and taxonomically positioned by full length 16S rRNA gene sequencing (see below, Table S2).

A set of 21 isolates representing four different major phyla (Table 1) with some redundancy were selected to assemble a synthetic soil bacterial community (SynCom). To prepare the SynCom inoculum, individual strains were plated from −80° stocks on PTYG agar and grown for 4 days at room temperature. Cells were then collected from the plates by washing with 5 mL of soil buffer (containing per L, 0.6 g of MgSO$_4$·7H$_2$O, 0.1 g of CaCl$_2$ and 1.8 mL of 5 x M9 minimal salts solution [BD Biosciences]). Individual cell suspensions were serially diluted in soil buffer and stained with SYBR green I for 15 min in the dark, according to manufacturer's instructions (Invitrogen), after which cell numbers were counted by flow cytometry (see below). Pure cultures were then diluted in soil extract (SE, see below) and mixed to obtain a suspension of in total $10^7$ cells per mL, and with approximate equal starting abundances of each individual member.

**Soil microcosm preparation.** Both NatComs and SynComs were cultured and passaged in sterile soil microcosms, based on a coarse silt supplemented with a sterile soil extract solution. The soil matrix was prepared from riverbank sediment (0 to 10 cm horizon) of the Sorge river sampled at the campus of the University of Lausanne (46°31'22.4N 6°34'31.7e), as previously reported (39). The material was transported to the laboratory, spread in a 5-cm layer in trays and air-dried in a ventilated hood at 23°C for 2 weeks, followed by double sieving to retrieve the 0.5 to 3 mm sized soil fraction. Sieved soil was divided in 2-kg portions, autoclaved for 1 h at 120°C (without the dry cycle) and dried for an additional 7 days as described above. Batches of soil (90 g for the first inoculation series, 80 g for subsequent transfers) were then distributed into 500-mL Schott borosilicate glass flasks with plastic screw cap and seal. Individual flasks with soil were again autoclaved (20 min, 120°C) to kill any remaining spores and vegetative cells. The sterility of the soil was confirmed after the second autoclaving by washing cells and spores from batches of 10 g of soil with 20 mL sterile 0.2% pyrophosphate solution, and plating serial dilutions on three different agar media: PTYG (see above), R2A (DSMZ GmbH), and Nutrient Agar (BD Biosciences). Absence of grown colonies after 3 weeks incubation at room temperature was taken as indication for the material to be sterile. All microcosms used in the study originated from the same batch of sieved soil.

As additional source of nutrients for all microcosms we produced a SE from the same soil as used for the NatCom and the SynCom isolates (see above), as follows. Top soil material (1 to 5 cm layer, 6 kg) was sampled as before and mixed in a 1:1 volumetric ratio with tap water in batches of 2 kg. The mixture was autoclaved (1 h, 120°C), mixed and left to settle overnight. The resulting supernatant was decanted into sterile 250-mL centrifuge tubes, centrifuged at 5,000 × *g* for 15 min to remove solids and pooled into 500-mL Schott flasks. This solution was autoclaved once more and then filtered through a 0.2-$\mu$m Stericup Quick Release System PES filter (Merck) into clean sterile Schott glass flasks and stored at room temperature in dark. The pH of SE was 5.28 ± 0.03. Its total organic carbon content (TOC) equaled 753 ± 49 mg C l$^{-1}$. A single batch of SE was used for all microcosms in this study.

**Soil microcosm inoculation and culturing.** Soil microcosms were inoculated with NatCom (four replicates) and SynCom (10 replicates) suspensions, and cultured either as a long-term single batch incubation, or through multiple 1-week growth and dilution cycles (Fig. 1A). Each microcosm initially comprised 90-g dry sterile soil matrix in a 500-mL screwcap glass bottle, amended with 10 mL community inoculum (at $10^7$ cells mL$^{-1}$ in SE, see above), thus resulting in ca. 10% gravimetric water content and $10^6$ cells g$^{-1}$ soil at start. The pH(H$_2$O) of the soil microcosms after inoculation was 8.62 ± 0.04. Uninoculated soils (four replicates) amended with 10 mL sterile SE served as controls for potential contamination. To contrast community growth in liquid suspension, the same SynCom and NatCom inocula were grown directly in 10 mL SE in 50 mL sterile Falcon tubes (starting at $10^7$ cells mL$^{-1}$), which were incubated at ambient temperature in the dark. After inoculation and before each sampling the soil microcosms were thoroughly mixed on a horizontal roller mixer (20 min at 80 rpm). SE liquid cultures were vortexed for 1 min every day.

In the long-term incubation series, samples (20 g) for community analysis (see below) were taken from each replicate microcosm after 1 week, 2 months, and 6 months. In the cycling regime, 11 g of the microcosm material were aseptically transferred after 1 week of growth to a fresh flask containing 80 g of dry sterile soil matrix. Again, 9 mL of sterile SE was added to maintain moisture content and replenish nutrient levels, thus resulting in 10-fold microcosm dilution upon each transfer. Flasks were again incubated for 1 week as before with intermittent roller-mixing. This incubation-dilution cycle was repeated eight times consecutively.

SE-liquid cultures were sampled (2 mL) each week for community analysis, after which 1 mL was transferred to a fresh tube with 9 mL of sterile SE. Incubation and dilution were repeated for eight cycles, similar as for the soil microcosms with the cycling regime. A further SE-liquid control was prepared for the long incubation (1 week, 2 months, and 6 months).

**Chemical perturbation.** In order to assess the effect of chemical perturbance on the resilience of the established communities, five of 10 SynCom replicates (both soil microcosms and SE-liquid) after the fifth transfer (see above) were exposed to toluene vapor during 1 week, as follows. After the inoculation with material from the previous cycle, heat-sealed 1 mL (for soil microcosms) or 0.2 mL (for SE-liquid) micropipette tips were placed inside the flasks, open at the top to the air, and filled with 100 $\mu$L or 10 $\mu$L pure toluene, respectively. These volumes are equivalent to a nominal concentration of 1.88 mM toluene, which will partition into the gas and aqueous phases in both systems. Microcosm flasks and tubes were tightly closed and incubated for 7 days with daily mixing (during each mixing, the toluene reservoir was briefly removed and then placed back). Samples were taken at day 7, and material from the exposed microcosms was again diluted as before into fresh soil microcosms or SE-liquid, but without toluene. The non-exposed growth regime was repeated for another two cycles to study community recovery.

**Community analysis.** Samples of 20 g (soil microcosms) or 2 mL (SE-liquid) were mixed with 20 mL of sterile pyrophosphate solution (see above) and vortexed for 1 min at maximum speed. The samples were left to stand for 1 min to settle soil particles, after which the supernatant was transferred aseptically to a new vial. An aliquot of 100 $\mu$L of each sample supernatant (containing the cell suspension) was mixed with an equal volume of 4 M sodium-azide solution to fix the cells. Fixed samples were kept at 4°C until flow cytometry counting (see below).

The rest of the supernatant cell suspension ($\sim$19 mL) was centrifuged in a swing-out rotor (Eppendorf A-4-62 Swing Bucket Rotor) at 3,200 $\times$ $g$ for 10 min to pellet cells. The liquid was discarded and cell pellets were frozen at $-80$°C until DNA isolation. Cell pellets were thawed and DNA was purified using a DNeasy PowerSoil kit (Qiagen) according to the manufacturer's protocol. The concentration of purified DNA was measured using a Qubit dsDNA BR assay kit (Invitrogen). DNA samples were stored at $-20$°C until library preparation (see below).

**Flow cytometry.** Cell suspensions were filtered using a 40-$\mu$m nylon cell strainer (Falcon) and then fixed (see above). Fixed cell suspensions were serially diluted in sterile saline and stained with SYBR green I for 15 min in the dark according to instructions of the supplier (Invitrogen). Stained cells suspensions were counted in 20 $\mu$L sample volume at medium flow rate (60 $\mu$L min$^{-1}$) using an ACEA NovoCyte Green flow cytometer (OMNI Life Science Agilent). The SYBR green I signal was measured in the FITC-channels of the instrument. Based on buffer controls, events with FSC-H-values above 50 and FITC-H above 350 were considered to potentially originate from microbial cells. Uninoculated microcosms, extracted and fixed in the same way, served to quantify cell-free (e.g., colloidal particles) background, which was subtracted from inoculated microcosm samples.

**Identification of soil isolates.** Each soil isolate was identified based on the near-full length 16S rRNA gene, amplified by PCR with Phusion U Hot Start PCR MasterMix (Thermo Fisher Scientific) in the presence of 0.5 mM betaine (Sigma-Aldrich) using universal bacterial primers (27F 5'-AGAGTTTGATCCTGGCTCAG and 1492R 5'-GGTTACCTTGTTACGACTT, or 27F_deg 5' AGRGTTYGATYMTGGCTCAG and 1391R_v18 5' GACGGGCGGTGWGTRCA) (64). Amplified DNA was purified using Gel and PCR Clean-up kits (Macherey-Nagel) and single-end Sanger-sequenced with the corresponding forward primer at Eurofins Scientific. Sequences were compared to the SILVA database (version 132) using BLAST (65) with default parameters for the genus level identification.

**Community 16S rRNA gene amplicon sequencing.** Aliquots of 10 ng purified DNA per sample were used to amplify the V3 to V4 region of the bacterial 16S rRNA gene, following the Illumina 16S Metagenomic Sequencing Library protocol (https://support.illumina.com/content/dam/illumina-support/documents/documentation/chemistry_documentation/16s/16s-metagenomic-library-prep-guide-15044223-b.pdf), indexed with a set A Nextera XT Index Kit (v2, Illumina), quantified and pooled in equal amounts for sequencing. The pooled SynCom amplicon libraries were spiked with 25% PhiX control DNA and paired-end sequenced on an Illumina MiniSeq instrument with the mid-output flow cell (Illumina). NatCom libraries and a sample of the SynCom starting inoculum were sequenced on a MiSeq platform with 300 cycles MiSeq v3 paired-end sequencing at the Lausanne Genomic Technologies Facility. Given their known reduced composition, for SynCom samples only the V4-end reads were used for analysis. Raw sequence reads were quality checked using FastQC 0.11.7 (66), then cleaned and trimmed where necessary using Trimmomatic 0.36 (67). Primer sequences, ends with low quality, and reads with poor quality score were removed. The quality was re-checked after trimming. A reference database of the inoculated SynCom members was created using the determined 16S rRNA gene sequences of each isolate (described above) and complemented by all unique sequence variants obtained from a MiSeq paired-end analysis of the SynCom inoculum. These reads were processed with QIIME 2 on a Unix platform (version qiime2-2018.8) (68), and grouped into taxonomic units at level 6 at 99% sequence identity by comparison to the SILVA database (version 132). Sequences were aligned using MUSCLE 3.8.1551 (69) and visualized using Jalview (70). Unique variable regions of 60- or 90-bp length were selected as identifier for each of the 21 SynCom strains. Strain abundances in the SynCom samples were then counted in the pools of quality-controlled sequence reads by searching for the unique selected sequence identifiers of each member in the reference database, using the bash command "grep." The obtained counts were corrected for the number of 16S rRNA operons in the respective SynCom isolates genomes (to be described elsewhere). Relative abundances were then normalized to the total number of classified reads in each sample, which was further

compared with differences in total cell count (as determined by flow cytometry) and the concentration of purified sample DNA.

**Microbe Atlas comparison.** All sample sequences were compared with a global background of soil communities from the Microbe Atlas Project database (MAPdb, https://microbeatlas.org). The raw 16S reads from all samples were standardized and quality-filtered using a custom C++ program employed internally by MAPdb and then mapped using MAPseq 1.2.6 (71) (reference database: MAPref v2.2; all other parameters kept at default) to obtain 97%-level OTU count tables compatible with MAPdb. Samples from MAPdb with meta-data annotations "soil" (main environment) or "rhizosphere" (sub-environment) were used for downstream analysis (110,928 samples total). Earth Microbiome Project (72) samples were identified based on accessions from https://ebi-metagenomics.github.io/blog/2019/04/17/Earth-Microbiome-Project/ and corresponding soil pH values were extracted via the "sample_ph" field from accession-matched Sequence Read Archive (73) annotation files.

**Simulation model.** To test the effects of stochastic variations in starting numbers of rapidly growing members within complex communities, we deployed a recently developed community model that simulates substrate-limited Monod growth of large numbers of bacterial taxa simultaneously (74). The model was seeded with 200,000 individual cells sampled with a weighted probability distribution from the measured relative abundances of 314 major taxa in a soil sample. Growth rates were attributed between 0.01 and 0.4 h$^{-1}$ according to the $\log_{10}$ relative taxa abundance at start, except for five taxa with subsampled starting numbers between 0 and 10 (of 200,000 cells in total) that were given growth rates of 0.55, 0.25, 0.8, 0.6, and 0.35 h$^{-1}$. Growth was allowed to proceed until all carbon was depleted, after which the final community was subsampled to 200,000 cells (to resemble a sequenced sample with $2 \times 10^5$ reads). Relative and stacked taxa abundances were plotted within these subsampled data sets. Simulations were repeated five times independently (Fig. S3).

**Statistical analyses.** Data were analyzed using R 3.6.1 (R Core Team, 2019) and the R packages *vegan* (75), *ggplot2* (76), *phyloseq* (77), *reshape* (78), and also using GraphPad Prism (version 9.0.0 for Mac OS X). The trends of microcosm total cell densities (as measured by flow cytometry) were compared using ANCOVA ($n = 4$ to 10 replicates per condition). Absolute abundances per SynCom community member were calculated from their relative (sequence) abundance times the measured total community size per replicate (from flow cytometry). The influence of culturing environment (e.g., soil, liquid) on community yield was compared using one-tailed t-tests. Differences in DNA yields were compared using a one-way ANOVA with *post hoc* Tukey's multiple-comparison test. The inter-replicate variability was expressed by the Bray-Curtis replicate distance from the community centroid. Effects of conditions were compared using one-way ANOVA with *post hoc* Tukey's multiple-comparison test. Alpha diversity was computed as community richness and Shannon indices. Communities at different time points and treatments were compared by NMDS using Bray-Curtis distance values of normalized relative community member abundances. Multivariate dispersion of the data was examined using the *betadisper* function from *vegan*. Adonis (MANOVA with 999 permutations) was used to assess the differences between groups based on the output of *vegdist* (Bray-Curtis distances). The effect of toluene exposure on community cell densities was assessed using a Wilcoxon matched-pairs signed rank test. The effect of time and toluene exposure on community richness and Shannon values was assessed using two-way ANOVA. Clustered heatmap and UMAP (79) projections were generated from Bray-Curtis distance matrices using julia 1.6.0 (80) and the *Distances.jl* package (81), (version 0.10.3). UMAP projections were computed using the *UMAP.jl* package (https://github.com/dillondaudert/UMAP.jl, version 0.1.8; parameters: n_neighbors = 500, min_dist = 1.5, spread = 15, epochs = 2,000). Scatterplots were produced using python 3.9.1 (82) and the *seaborn* package (83), (version 0.11.0).

**Analysis of soil parameters.** The gravimetric water content in twice autoclaved soil was determined from weight loss of soil samples before and after drying at 70°C for 10 days. Soil-pH was measured in mixed solution with distilled water, stirred for 1 h at 120 rpm, using an Orion Star A111 Benchtop pH Meter (Thermo Fisher Scientific).

Organic material was characterized by UV/Vis and fluorescence spectrometry. Soil-water (5 g) or SE-water (5 mL) extracts were prepared by mixing sample in 12 replicates with 20 mL MilliQ water at 80 rpm for 1 h. Mixtures were subsequently centrifuged for 15 min at 4,600 $\times$ *g* and the supernatant was filtered using a 0.2-$\mu$m Stericup Quick Release System PES filter (Merck). Filtered samples were stored in glass amber vials at 4°C in the dark prior to analysis. Filtered samples were then serially diluted in MilliQ water, transferred to 1-cm quartz cuvettes and measured in a UV/Vis spectrophotometer (Perkin Elmer 650S) or a Fluorolog-3 spectrofluorometer (Horiba). Data were collected in the three-dimensional form of excitation-emission matrices (EEMs) for a parallel factor analysis (PARAFAC) model, against MilliQ water. Excitation wavelengths ranged from 270 to 500 nm and emissions were measured in the range from 300 to 600 nm. Data were processed using the PARAFAC algorithms (84) in MATLAB (vs.2016a, MathWorks). Detected spectra correspond to six different organic matter types as described by Fellman et al. (85), although more recent categorization would follow Lehmann and Kleber (86). $NH_4$-N, $NO_3$-N and total-N in the final soil+SE was determined by Sol-Conseil (Gland, Switzerland).

**RockEval methodology.** RockEval analysis was used to assess the carbon content composition of natural soil, autoclaved soil and soil mixed with SE, as suggested (87). Upon mixing and drying to remove the remaining water content, the samples were grounded using a Planetary Micro Mili Pulverisette 7 (Fritsch). The samples (including the IFP160000 standard) were processed using a RockEval 6 Pyrolyser (Vinci Technologies) at the Faculty of Geosciences and Environment, University of Lausanne. In short, samples were pyrolysed and combusted, leading to the release of hydrocarbons (S$_1$ peak), kerogen (S$_2$), and CO$_2$ (S$_3$), and remainder residual carbon (RC), which were measured by flame ionization and thermal conductivity detectors (87). The obtained values of S$_1$, S$_2$, and S$_3$ were used to calculate the total organic carbon (TOC),

pyrolyzable and mineral carbon fractions, and the hydrogen (HI) and oxygen indices (OI). HI represents the ratio of hydrogen to organic carbon and is indicative of the origin of the organic material. OI shows the amount of oxygen relative to TOC. These indices are calculated as follows:

$HI = S_2/TOC \times 100;$

$OI = S_3/TOC \times 100$

TOC of the SE-solution was determined by Scitec Research SA (Lausanne, Switzerland).

**Data availability.** The NatCom and SynCom 16S rRNA gene amplicon sequencing data are available from the Short Read Archives under BioProject number PRJNA767350.

## SUPPLEMENTAL MATERIAL

Supplemental material is available online only.

**FIG S1**, TIF file, 1 MB.
**FIG S2**, TIF file, 0.5 MB.
**FIG S3**, TIF file, 1.2 MB.
**FIG S4**, TIF file, 7.5 MB.
**FIG S5**, TIF file, 0.9 MB.
**FIG S6**, TIF file, 0.6 MB.
**FIG S7**, TIF file, 1.1 MB.
**TABLE S1**, DOCX file, 0.04 MB.
**TABLE S2**, DOCX file, 0.1 MB.

## ACKNOWLEDGMENTS

The authors acknowledge the support of the Lausanne Genomics Technology Facility for the amplicon sequencing, and Michael Rowley, Thibault Lambert, and Thierry Adatte from the Faculty of Geosciences and Environment, University of Lausanne, for their help in soil analysis.

This work was supported by the Swiss National Centre in Competence Research *NCCR Microbiomes* (No. 51NF40_180575).

S.C., V.S., and J.vdM. conceptualized the work; S.C. and V.S. carried out experiments; S.C., V.S., J.T., C.vM., and J.vdM. analyzed data; S.C. and J.vdM. prepared the draft manuscript; J.vdM. and C.vM. provided funding. All authors approved the final manuscript.

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
