## [Reviewer comments · mSystems]

Reproducible Propagation of Species-Rich Soil Bacterial Communities Suggests Robust Underlying Deterministic Principles of Community Formation

Senka Causevic, Janko Tackmann, Vladimir Sentchilo, Christian von Mering, and Jan van der Meer

Corresponding Author(s): Jan van der Meer, University of Lausanne

Review Timeline:

Submission Date:

February 20, 2022

Accepted:

March 14, 2022

Editor: Sarah Allard

Reviewer(s): The reviewers have opted to remain anonymous.

Transaction Report:

DOI: <https://doi.org/10.1128/msystems.00160-22>

March 14, 2022

Prof. Jan Roelof van der Meer
University of Lausanne
Department of Fundamental Microbiology
Batiment Biophore
Quartier Unil-Sorge
Lausanne CH 1015
Switzerland

Re: mSystems00160-22 (Reproducible Propagation of Species-Rich Soil Bacterial Communities Suggests Robust Underlying Deterministic Principles of Community Formation)

Dear Prof. Jan Roelof van der Meer:

The resubmission comprehensively addresses the reviewer concerns and is much improved. Reviewer 1 was able to review this version and agrees that the comments have been sufficiently addressed.

Your manuscript has been accepted, and I am forwarding it to the ASM Journals Department for publication. For your reference, ASM Journals' address is given below. Before it can be scheduled for publication, your manuscript will be checked by the mSystems production staff to make sure that all elements meet the technical requirements for publication. They will contact you if anything needs to be revised before copyediting and production can begin. Otherwise, you will be notified when your proofs are ready to be viewed.

Publication Fees:

We recognize that the video files can become quite large, and so to avoid quality loss ASM suggests sending the video file via <https://www.wetransfer.com/>. When you have a final version of the video and the still ready to share, please send it to mSystems staff at mssystemsjournal@msubmit.net.

For mSystems research articles, if you would like to submit an image for consideration as the Featured Image for an issue, please contact mSystems staff at mssystemsjournal@msubmit.net.

Sincerely,

Sarah Allard
Editor, mSystems

Journals Department
FIG S5: Accept
Table S1: Accept
FIG S7: Accept
FIG S3: Accept
FIG S6: Accept
FIG S1: Accept
Table S2: Accept
FIG S2: Accept
FIG S4: Accept